# Diversity and Phosphate Solubilizing Characteristics of Cultivable Organophosphorus-Mineralizing Bacteria in the Sediments of Sancha Lake

**DOI:** 10.3390/ijerph19042320

**Published:** 2022-02-17

**Authors:** Yong Li, Xintao Yu, Jiarui Zheng, Zhilian Gong, Wenlai Xu

**Affiliations:** 1Faculty of Geosciences and Environmental Engineering, Southwest Jiaotong University, Chengdu 610059, China; yuxintao@my.swjtu.edu.cn (X.Y.); zjr285829@my.swjtu.edu.cn (J.Z.); 2School of Food and Biological Engineering, Xihua University, Chengdu 610039, China; 3State Key Laboratory of Geohazard Prevention and Geoenvironment Protection, Chengdu University of Technology, Chengdu 610059, China; xuwenlai2012@cdut.cn

**Keywords:** sediments, organophosphorus-mineralizing bacteria (OPB), diversity, phosphorus-solubilizing characteristics

## Abstract

The organophosphate-mineralizing bacteria (OPB) convert environmental organic phosphorus (P) into soluble P that can be directly absorbed and utilized by organisms. OPB is an important group of microorganisms in lake sediments. The P decomposed and released from the sediments by OPB is an important P-source in eutrophic water bodies. In this study, the egg-yolk organophosphate medium was used to isolate and screen OPB strains from the sediments of Sancha Lake. Furthermore, the obtained OPB strains were classified based on their 16S rDNA sequence. Both the solid and liquid lecithin hydrolyzing experiments were conducted to investigate the P-solubilizing characteristics of the obtained OPB strains. Microcosm experimentsiwere performed to study the P-release ability of OPB strains from sediments. A total of 39 OPB strains were isolated from the sediments of Sancha Lake. They belonged to three phyla, five families, and five genera, and contained two potentially new species. *Bacillus* and *Pseudomonas* were the dominant genera. On the solid lecithin plate, 35 of the 39 OPB strains produced visible phosphate halos, and 24 strains showed a high ratio of P halo diameter (HD)/colony diameter (CD). In the liquid lecithin medium, all 39 OPB strains demonstrated P-solubilizing ability, but with significant differences. The *Pseudomonas* strain demonstrated the strongest P-solubilizing ability, at 70.91 mg·L^−1^. There was no significant correlation between the amount of released phosphorus by OPB strains and pH. The P-solubilizing characteristics of OPB were affected by the interaction of dissolved inorganic phosphate and alkaline phosphatase. In the microcosm experiments, the added OPB strains significantly promoted the decomposition and release of organic phosphorus (OP) in the sediments. OPB in the sediments of Sancha Lake is rich in diversity and had a strong ability to release OP in the sediments.

## 1. Introduction

When external input of phosphorus (P) is under effective control, the release of endogenous phosphorus from lake sediments significantly affects the increase in its concentration in water bodies [1]. The elevated P concentration in lake water may lead to eutrophication, thereby destroying the aquatic ecosystem [2]. Therefore, it is particularly important to study the release capacity of endogenous phosphorus in sediments and its influencing factors. According to the European Commission’s Standards, the Measurements and Testing Programme (SMT), the endogenous phosphorus (total phosphorus, TP) in the sediments comprises organic phosphorus (OP) and inorganic phosphorus (IP). The release of endogenous P in sediments was found to be significantly affected by microorganisms, rather than by physicochemical effects [3]. Phosphate solubilizing bacteria in sediments are a kind of functional bacteria, which can convert phosphate not being used by phytoplankton into available dissolved phosphorus. The bacteria can be divided into inorganic phosphate solubilizing bacteria and organophosphorus-mineralizing bacteria (OPB) according to their phosphorus sources. OPB produce extracellular alkaline phosphatase (ALP), which hydrolyzes and mineralizes organophosphate [4], and generates dissolved inorganic phosphorus (DIP) that can be absorbed by algae to subsequently cause eutrophication [5]. OPB can increase the concentration of bioavailable phosphorus in the water body, and the OPB are an important group of microorganisms in sediments. They play a vital role in the P-cycle of eutrophic water bodies [6]. Therefore, the study of OPB can help to deeply understand the role of microorganisms in the process of phosphorus release from sediments.

OPB exists widely in the environment. The current research reports on cultivable OPB primarily focus upon soil [7]. Some scholars studied the number, type, and distribution of OPB in oceans, rivers, and lakes [8,9,10,11]. However, there have been only a few studies on the types and characteristics of cultivable OPB and their potential to release phosphorus in sediments in the eutrophic lake ecosystem.

The Sancha Lake is an important source of drinking water in the eastern new district of the Sichuan Province. It has a serious eutrophication problem [12]. Our previous study showed that the sediments of the Sancha Lake are rich in OP, which is an important form of TP [13]. Under the controlled external input of P, its release from the sediments of the Sancha Lake became an important replenishment source of phosphorus in the overlying water [14]. Currently, there is a lack of research on the cultivable OPB diversity in the sediments of the Sancha Lake and their ability to release P from the sediments. Therefore, isolating and screening OPB strains from the sediments of Sancha Lake, classifying them, and analyzing their P-solubilizing characteristics and P-release ability, is important for understanding the pattern of OP release from the sediments of Sancha Lake and restoring eutrophication.

We hypothesized that: (1) There are abundant and diverse OPB in the sediments of Sancha Lake and (2) OPB have a strong ability to release OP from the sediments.

## 2. Materials and Methods

### 2.1. Overview of Sancha Lake and Collection of Sediments

The Sancha Lake is located in the eastern new district, Sichuan Province, China, 104°11′16″ E~104°17′16″ E, 30°13′08″ N~30°19′56″ N, the eastern suburbs of Chengdu, and the upper reaches of Jiangxi River, a tributary of the Tuojiang River in the Yangtze River system. There is 27 km^2^ area in the Sancha Lake, with a drainage area of 161.25 km^2^ above the dam. The average water depth is 8.3 m, and the deepest value is 32.5 m. The primary water source of the Sancha Lake is the Min River, which accounts for about 80% of the total water volume of the reservoir. The other 20% comes from rainfall and the two streams of the Tiaodeng River and the Longyun River. The Sancha Lake area has a humid subtropical monsoon climate, with an annual average temperature of 15.2~16.9 °C, no freezes in all seasons, and average annual precipitation of 786.5 mm.

Since 2015, the pH, ammonia nitrogen, chemical oxygen demand (COD_Mn_) and biochemical oxygen demand (BOD_5_) indexes of the surface water of the Sancha Lake have met the Class III requirements of “Surface Water Environmental Quality Standard” (GB3838-2002) in China, but the high TN and TP contents failed the requirements [15]. The TP range in the sediments of the Sancha Lake was 0.243~3.774 mg·g^−1^, with an average value of 1.670 mg·g^−1^, and the TN range was 1.325~2.615 mg·g^−1^, with an average value of 1.931 mg·g^−1^ [15]. The range of DO (dissolved oxygen) at the sediment–water interface of the Sancha Lake was 4.1~12.1 mg·L^−1^, with an average value of 7.1 mg·L^−1^ [15].

In May 2020, in each of the five functional divisions of Sancha Lake (Figure 1), with a depth of 19.0 m, 30.0 m, 26.0 m, 13.0 m, and 4.0 m, the lake-bed surface sediments were collected using a Peterson grab dredger, stored in a Ziploc bag, sealed, kept on ice, and transported immediately to the lab [14]. Simultaneously, at each sampling point, the overlying water at the sediment surface was collected by an airtight water sampler and transported to the lab in a timely manner following the storage and transportation requirements of the “Water Environment Monitoring Specification (SL219-98)”. The water sample was filtered through a 0.45 μm membrane filter and stored for later use.

### 2.2. Isolation and Screening of OPB

Ten grams of wet weight sediment was transferred into a 250 mL Erlenmeyer flask which contained 90 mL of 0.85% sterile saline, shaken at 25 °C and 160 rpm for 0.5 h to make a sediment suspension. The 10-fold dilution method was used to prepare suspension dilutions up to 10^−6^. At the same time, fresh sediment samples were weighed to determine water content. The phosphate solubilizing microorganisms were screened using a traditional dilution plate separation method [16]. One hundred microliters of each suspension dilution were spread on an egg-yolk agar (EYA) plate. The composition of EYA was D-glucose 10 g, (NH_4_)_2_SO_4_ 0.5 g, NaCl 0.3 g, MgSO_4_·7H_2_O 0.3 g, CaCO_3_, 5 g, FeSO_4_·7H_2_O 0.36 g, MnSO_4_·H_2_O 0.03 g, KCl 0.3 g, agar 20 g, and egg-yolk emulsion 50 mL with distilled water to 1000 mL, pH 7.0. For preparing egg yolk emulsion, the eggshell was cleaned with 75% alcohol wipes and broken at one end. The egg white was removed, the egg yolk was poured into a sterilized flask and mixed aseptically with an equal volume of sterile 0.85% saline [17]. Three replicates were prepared for each dilution. Bacterial colony forming units (CFU) were counted after three days incubation at 25 °C. Select the concentration gradient of colony number within 30~300 for calculation. The single colonies with transparent phosphate halos were picked and purified for five rounds or more to screen out strains with unstable P-solubilizing ability. The obtained single colonies were then transferred to Luria-Bertani (LB) slant medium (NaCl 5.0 g, peptone 10 g, beef extract 3 g, agar 15.0 g, added with distilled water to 1000 mL, pH 7.0), and stored in a 4 °C refrigerator.

### 2.3. 16S rDNA Amplification, Sequencing, and Phylogenetic Analysis

The total bacterial DNA was extracted with the bacterial genomic DNA extraction kit from Tiangen Biotech (Beijing) Co., Ltd., following the kit instructions. Then, the extracted bacterial DNA was taken as a template, the universal primers 27F (5’-AGAGTTTGATCCTGGCTCAG-3’) and 1492R (5’-GGTTACCTTGTTACGACTT-3’) were used to amplify the bacterial 16S ribosomal DNA (rDNA) sequences with the polymerase chain reaction (PCR) [18]. The PCR was carried out in a BioRad T100 Thermal Cycler.

The PCR reaction system (20 µL) was composed of 10 × Ex Taq buffer 2.0 µL, 5 U Ex Taq polymerase 0.2 µL, 2.5 mM dNTP mix 1.6 µL, 27 F 1 µL, 1492R 1 µ, DNA 0.5 µL, and ddH2O 13.7 µL. PCR reaction conditions were as follows: denaturation at 95 °C for 30 s, annealing at 56 °C for 30 s, extension at 72 °C for 90 s, for a total of 25 cycles, with initial denaturation at 95 °C for 5 min before the cycle, and extension at 72 °C for 10 min after the cycle. Each treatment was repeated three times. The PCR products were detected by 0.8% agarose gel electrophoresis. Then, the PCR products were purified by Axygen^®^ AxyPrep DNA Gel Extraction Kit (Axygen Biosciences, Union City, CA, USA) and followed by sequencing with a 3730xl DNA Analyzer conducted by Shanghai Majorbio Bio-pharm Technology Co., Ltd. (Shanghai, China).

The contaminated sequences from the raw bacterial 16S rDNA sequence data were removed using SeqMan II (DNASTAR Inc, Madison, WI, USA). Furthermore, the obtained data were curated to remove regions with low-quality nucleotide scores (i.e., ambiguous bases) and assembled into contigs. The obtained entire 16S rDNA sequence was sent for BLAST alignment at EzBioCloud (https://www.ezbiocloud.net accessed on 31 October 2021) to determine the similarity and related species of the strain. Then, similar and effectively published sequences of typical strains were retrieved from the database. After multiple sequence alignment by clustalX, sequence editing was performed. MEGA 7.0 was used to estimate the evolutionary distance using the Kimura 2-parameter (K2P) model. The neighbor-joining (NJ) method was used to construct the phylogenetic tree of the 16S rRNA gene sequence of the strains with a bootstrap value of 1000 [19].

### 2.4. Evaluation of P-Solubilizing Ability of the OPB Strains

#### 2.4.1. Determination of P-Solubilizing Ability by Solid Lecithin Plate

The OPB obtained by isolation and purification was spot-inoculated on a solid lecithin medium (D-glucose 10 g, (NH_4_)_2_SO_4_ 0.5 g, NaCl 0.3 g, MgSO_4_·7H_2_O 0.3 g, CaCO_3_ 5 g, FeSO_4_·7H_2_O 0.36 g, MnSO_4_·H_2_O 0.03 g, KCl 0.3 g, agar 20 g, added with distilled water to 1000 mL, pH 7.0. After autoclaving, this was cooled to 50 °C, and 3 g phosphatidylcholine was added). The plates were cultured at 28 °C for 3 days. The phosphate halo diameter (HD) and colony diameter (CD) were measured, and the value of HD/CD was calculated. The P-solubilizing ability of a strain was preliminarily determined by whether it exhibited a phosphate halo. The criterion for selecting colonies is a value of HD/CD > 1. HD/CD > 1.5 indicated a strong P-solubilizing ability [20]. Each treatment was repeated three times.

#### 2.4.2. Determination of P-Solubilizing Ability by Liquid Lecithin Medium

Each of the isolated OPB strains was inoculated to LB medium and cultured at 28 °C and 160 rpm for 24 h. The bacteria were collected from LB liquid medium and then were washed with sterile water. The bacteria above were resuspended in sterile water, prepared in a bacterial suspension (10^8^ CFU mL^−1^). Then, 2 mL of the bacterial suspension was transferred into 200 mL of liquid lecithin medium (same as solid lecithin medium except the agar) and cultured at 28 °C, 160 rpm for 108 h. The culture was sampled every 12 h to determine the DIP concentration, ALP activity, and pH. The P concentration was determined by the malachite green-phosphomolybdenum heteropoly acid spectrophotometric method [21]. The ALP activity of liquid lecithin medium was determined according to Chen Wang’s method [22]. The culture pH was determined with a PHS-3C pH meter. The CFU mL^−1^ by drop plate method were determined in lecithin medium [16]. For the above-mentioned experiments, a sample with no bacteria inoculated was set as the control and each treatment was repeated three times.

The reported DIP concentration and ALP activity values were adjusted by subtracting the control values. Enzyme activity was defined as the amount of enzyme that generates 0.01 µmol of free phenol per liter of culture per minute at 37 °C is defined as one unit (U).

#### 2.4.3. Microcosm Setup and P-Release by Bacterial Strains

The collected fresh sediments were subjected to continuous irradiation under Co-60 γ-ray with an energy of 20 kGγ/h for 12 h to kill all the microorganisms in the sediments. Then, 200 g of sterilized sediment was transferred into a 1-L ground stopper reagent bottle, mixed with 800 mL of the filtered overlying water, and 10 mL of the bacteria suspension of the representative OPB strain with an approximate value of 10^8^ CFU mL^−1^**.** A total of 15 parallel samples were set for each group. A total of five groups were set, including four groups of representative strains and one group of control with no bacteria. The treated samples were cultured at 13 °C (the temperature of the Sancha Lake mud–water interface) and 80 rpm for 5 days, and manually shaken twice a day [5]. Each day, three bottles from each group were taken out, NH_4_Cl was added to a final concentration of 0.8 mol·L^−1^, shaken for 30 min, and then centrifuged at 3500 rpm for 10 min. The supernatant was filtered using a 0.45 µm membrane filter. The Standards, Measurements, and Testing (SMT) protocol was followed to extract various forms of P in the sediments after centrifugation [23,24]. The malachite green-phosphomolybdenum heteropoly acid spectrophotometric method was used to determine the content of various forms of P in the supernatant filtrate and sediments [21].

#### 2.4.4. Statistical Analysis

Statistical analysis was performed using SPSS statistical software (version 20.0, IBM, Armonk, NY, USA). The one-way analysis of variance (ANOVA) was used to analyze the differences of HD/CD, DIP, OP, ALP and pH among different OPB strains. Significance levels were set at *p* = 0.05 in all statistical analyses.

## 3. Results and Discussion

### 3.1. Results of OPB Screening and Identification

In this study, a total of 39 OPB strains with stable P-solubilizing activities were isolated from the sediments of the Sancha Lake. They belonged to three phyla, including Firmicutes, Proteobacteria, and Bacteroidetes; five families, including *Bacillaceae*, *Aeromonadaceae*, and *Comamonadaceae*; five genera, including *Bacillus*, *Pseudomonas*, and *Aeromonas*; and 22 species including *B. thuringiensis*, *B. altitudinis*, and *B. mobilis*. The details are listed in Table 1 and Figure 2. Common OPB genera include *Bacillus*, *Pseudomonas*, *Acinetobacter*, and *Aeromonas* [4,25,26]. Our study also demonstrated that *Bacillus* and *Pseudomonas* as the dominant OPB genera in the sediments of the Sancha Lake, whereas *Acidovorax* was rare. According to the established strain identification method [27], strains SWSO1719 and SWWO1721 were potentially new species. There are a large number of OPB strains in the sediments of the Sancha lake, of which the number of OPB strains is 3.17 × 10^6^ CFU g^−1^ (dry sediment). The number of isolated OPB species and their P-solubilizing activities in this study were both higher than those from the Dutang reservoir sediments reported by Xuejin You et al. [22]. Compared with the OPB strains obtained by Wenhong Li et al. [28] from the sediments of Xiaonan Lake in West Lake, Hangzhou, the OPB strains isolated in this study exhibited slightly lower P-solubilizing activities; nevertheless, with higher species richness. The sediments of the Sancha Lake exhibited rich diversity and new taxa in the cultivable OPB. These results thus enrich the existing knowledge on species of lake OPB and expand the source of P-dissolving microorganisms. We used only the 28 °C and pH 7 in isolation. They could have had limited the bacteria which we could isolate. The effects of different temperature and pH on OPB isolation need to be further studied. 

### 3.2. P-Solubilizing Ability

#### 3.2.1. P-Solubilizing Activity of Screened OPB Strains

The HD/CD values of the 39 OPB strains are shown in Table 2. According to the threshold of HD/CD [20], four strains of OPB exhibited no P-solubilizing ability, 11 strains had weak P-solubilizing ability, and 24 strains exhibited strong P-solubilizing ability. Among the 35 OPB strains with phosphate halos, the HD/CD values of SWSO1719, SWSO172, SWSO173, SWSO174, SWSO175, and SWSO1712 were significantly higher than those of the other strains (*p* < 0.05). Among them, the highest value was from SWSO1719. The HD/CD values of SWWO1711, SWWO1713, SWWO1714, SWSO1718, and SWWO1721 were significantly lower than those of other strains (*p* < 0.05). Among them, the lowest value was from SWWO1711.

The P-solubilizing results of the 39 OPB strains by liquid lecithin experiment are mentioned in Table 2. All the media inoculated with OPB strains turned from turbid to clear. After subtracting the value from control, the DIP concentration was 0.47~70.91 mg·L^−1^. The best P-solubilizing result was shown by SWSO178, with a DIP concentration of 72.26 mg·L^−1^, an increase of 70.91 mg·L^−1^ compared to that of the control. The increase in DIP concentrations of SWWO175, SWWO1716, and SWWO1717 was not significant (*p* > 0.05). SWWO1717 exhibited the weakest P-solubilizing effect, with a DIP concentration of 1.82 mg·L^−1^, only 0.47 mg·L^−1^ higher than that of the control.

SWWO175, SWWO1718, SWWO1719, and SWWO1720 exhibited no phosphate halos on the solid lecithin medium, but they displayed P-solubilizing ability in the liquid lecithin experiment. Among them, the DIP concentration in SWWO1718 culture reached 49.21 mg·L^−1^. As we see, the quantitative liquid lecithin experiment and the qualitative solid lecithin experiment gave similar results in the determination of the P-solubilizing ability, but there were certain differences as well. The P-solubilizing substance (such as ALP) produced by bacterial metabolism in the solid plate probably did not diffuse or did not diffuse enough.

According to Table 2, all the 39 OPB strains screened from the sediments of the Sancha Lake could produce ALP, but the ALP activity was not consistent with the P-solubilized amount. For example, SWSO178 and SWSO1719 could dissolve more P, but showed lower ALP activity. This may be because only the maximum DIP value and the ALP activity of a strain were compared, without considering their dynamic changes.

#### 3.2.2. Changes in Alkaline Phosphatase and pH of the Organophosphate Culture of the Four Representative Organophosphate-Mineralizing Bacteria Strains and Their Organophosphate Solubilizing Ability

To further explore the P-solubilizing mechanism of OPB, one representative OPB strain with the highest activity was selected from each of the dominant genera, including *Bacillus*, *Pseudomonas*, and *Aeromonas.* In addition, one representative strain was selected from the genus *Acidovora**x* which was rarely reported with P-solubilizing characteristics. The growth dynamics of these four representative OPB strains were investigated in the liquid lecithin medium. Furthermore, the impact of changes in DIP, ALP, and pH on the P-solubilizing ability of these strains was explored.

As shown in Figure 3, the DIP concentration and ALP activity of the four OPB strains in the liquid lecithin culture were significantly higher (*p* < 0.05) than that in the control. Among these, SWSO178 and SWSO1719 exhibited strong P-solubilizing ability, while SWWO1713 and SWWO1711 showed weak P-solubilizing ability. The P-solubilizing activity of SWSO178 and SWSO1719 reached a peak at 48 h, while that of SWWO1713 and SWWO1711 reached a peak at 72 h. The DIP concentration changes of the four OPB cultures presented different curves, nevertheless, the curves all had a trough–peak–trough pattern.

The curves of changes in ALP activity of the four OPB strains also showed different trends in the liquid lecithin culture. The ALP activity of SWSO178 and SWSO1719 reached a peak at 36 h, showing a stronger enzymatic activity, while that of SWWO1713 and SWWO1711 reached a peak at 72 h, indicating a weaker enzyme activity. Considering the relationship between the DIP and ALP curves, DIP concentration increased with the increase in the ALP activity, but the DIP changes and peaks lagged in the liquid lecithin culture.

The P-solubilization of OPB was due to the enzymatic hydrolysis of ALP produced by bacteria [29]. In the liquid medium using lecithin as the organophosphate source, the DIP curve had a pattern of “trough-peak-trough”. This was mainly due to the interaction between DIP and ALP. Our result was consistent with that of Rossolinia et al. [30].

ALP is a non-specific phosphomonoesterase [31] and an inducible enzyme. The synthesis and activity of ALP are determined by DIP concentrations in the surrounding environment [32]. ALP is also positively correlated with OPB biomass [33]. In this study, the concentration of DIP increased with the increase in ALP. The activity of ALP peaked first, followed by the DIP concentration. When the bacterial cells entered the exponential phase, their growth, reproduction, and metabolism were vigorous, and promoted ALP production, resulting in elevated DIP concentration. When the environmental DIP concentration was too high, the activity of ALP was inhibited. As the bacteria approached the end of exponential growth, they entered the phosphorus starvation state, and the inhibition of ALP was reversed. Ecologically, this means that OPB produce ALP under the phosphorus-deficient condition to decompose organophosphorus, meeting their growth needs [34]. Such characteristics of change are called “inhibition–uninhibition” [35] or “induction–inhibition” mechanisms [36]. This is widely observed in rivers, reservoirs, ponds, and oceans [34]. In addition to the related to enzymatic activity, OPB growth and reproduction could have impact on DIP concentration [7]. In this study, the relationship of OPB biomass and DIP concentration need to be further studied.

As shown in Figure 4, the pH of SWSO178 culture dropped from 7.3 to 6.0 at 36 h of culture, stabilized at 6.0–6.2, and began to increase at 72 h, peaking at 7.3 at 96 h. At 108 h, the pH dropped to 7.0, lower than that of the control. The pH of the SWSO1719 culture dropped to the lowest at 36 h, decreasing from 7.3 to 6.2, then began to increase at 48 h to 7.5 until 84 h, and, at 108 h, dropped to 7.1, slightly lower than that of the control. The pH of the SWWO1713 was 6.3, the lowest at 48 h after inoculation, increased to the maximum value of 7.8 at 84 h, and then began to decline. The pH of the SWWO1711 culture reached the trough at 48 h, then began to increase at 60 h to reach 8.1 at 84 h, and further reduced to 7.5 at 108 h, but was higher than that of the control.

The culture pH of the four representative OPB strains all showed a “trough–peak–trough” pattern. Figure 3 and Figure 4 show that pH decreased with the increase in culture DIP concentration, and pH increased when the decrease in culture DIP concentration. We observed a certain correlation between the DIP concentration and pH in OPB culture, although, this correlation was not significant (*p* > 0.05). The drop in pH may be because the cells entered the exponential phase, with vigorous growth, reproduction and metabolism, and increased ALP production. ALP thus promoted the hydrolysis of lecithin into DIP and organic acids. On entering the stationary phase, microbial cell metabolism slows down, respiration becomes weak, and cells begin to lyse. The choline produced by ALP to decompose lecithin in this study could not be completely hydrolyzed, leading to an increase in pH. The decline in pH of the culture is due to the production of organic acids or lecithin degradation products of OPB [37], or there may be other sources of H^+^, such as the NH_4_^+^/H^+^ exchange and respiration. However, the decrease in the culture pH is not a necessary condition for phosphorus solubilization [38]. In fact, the correlation between pH and DIP concentration in OPB culture is weak [7].

#### 3.2.3. P-Release by OPB Strains in Sediments

The above-mentioned four representative OPBs were used for the microcosmic sediment P-release experiment. As shown in Figure 5, the added strain culture had various effects on the release of sediment P, but all exhibited significantly higher release of P than the control (*p* < 0.05). The DIP concentration of the overlying water in the control was stabilized at approximately 0.17 mg·L^−1^ after three days. In the SWSO178 and SWSO1719 cultures, the DIP concentration of the overlying water changed in a trough–peak–trough manner. In detail, the overlying water DIP concentration of SWSO178 culture was at the trough on day 1, peaked on day 3 at 0.952 mg·L^−1^, and then decreased, but was still higher than the control at the end of the experiment. The overlying water DIP concentration of SWSO1719 culture was the lowest on day 2, increased the maximum to 0.714 mg·L^−1^ on day 3, which was 4.2 times higher than that of the control, followed by a decrease on days 4 and 5, but were still higher than that of the control. In SWWO1713 and SWWO1711 cultures, the overlying water DIP changed in a down–up way. In detail, the overlying water DIP increased slowly on day 1 peaked on day 4, at 0.544 mg·L^−1^ and 0.425 mg·L^−1^, respectively, and the values were, respectively 2.6 and 2.4 times compared with the control, and then reduced on day 5, but both concentrations were still higher than that of the control.

As shown in Figure 5, compared with the control, the sediment OP contents in the four OPB cultures reduced significantly (*p* < 0.05). SWSO178, SESO1719, SWWO1711, and SWWO1713 can increase DIP content in overlying water and reduce OP content in sediment, which may be related to the ALP secretion by the strains. After adding the SWSO178, SWSO1719, SWWO1711, and SWWO1713 strains, the sediment OP contents reduced by 14.5%, 13.1%, 8.6%, and 7.3%, respectively. The decrease in sediment OP in SWWO1711 and SWWO1713 cultures was consistent with the increase in DIP concentration of the overlying water, whereas in the SWSO178 and SWSO1719 cultures, the sediment OP decreased; in fact, the overlying water DIP also decreased in the early stage, but in the middle and later stages, there was a consistent decrease in the sediment OP and increase in the overlying water DIP. This may be because SWSO178 and SWSO1719 grew too vigorously in the early stage, and the soluble P released from sediment OP was not sufficient, therefore, the cells supplemented from the soluble P in the overlying water to meet their biological P needs. Baiwen Hu et al. [39] came to a similar conclusion in their research on the P released from sediments by OPB in a simulated pond environment.

The sediments of Sancha Lake are rich in OP, which is an important source of P in the water body. Under favorable environmental factors, high-efficiency OPB can convert OP in sediments that are insoluble or not directly usable by organisms into DIP and released into water bodies. Therefore, the upper water body of the Sancha Lake is at a risk of more serious eutrophication.

## 4. Conclusions

A total of 39 OPB strains were isolated from the sediments of the Sancha Lake. These strains were found to belong to three phyla, five families, and five genera. *Acidovorax* has been rarely reported as OPB, while SWSO1719 and SWWO1721 were potentially new taxa. *Bacillus* and *Pseudomonas* were the dominant genera. The results show that the cultivable OPB in the sediments of Sancha Lake have a rich diversity and some new taxa, expanding the source of P-solubilizing microorganisms.

All the 39 OPB strains could dissolve lecithin, but the OPB strains showed significantly different P-solubilizing ability. The primary mechanism of P-solubilization by OPB may be through the production of ALP. The correlation between the amount of P-released and pH was not significant. The P-solubilization characteristics of OPB were affected by the interaction between the DIP concentration and the ALP activity in the culture medium.

The OPB strains could significantly promote the mineralization and decomposition of organophosphorus compounds in the sediments of the Sancha Lake. OPB play an important role in the decomposition and release of OP in the sediments of Sancha Lake. The P decomposed and released from the sediments by OPB may be an important P-source in eutrophic water bodies.

## Figures and Tables

**Figure 1 ijerph-19-02320-f001:**
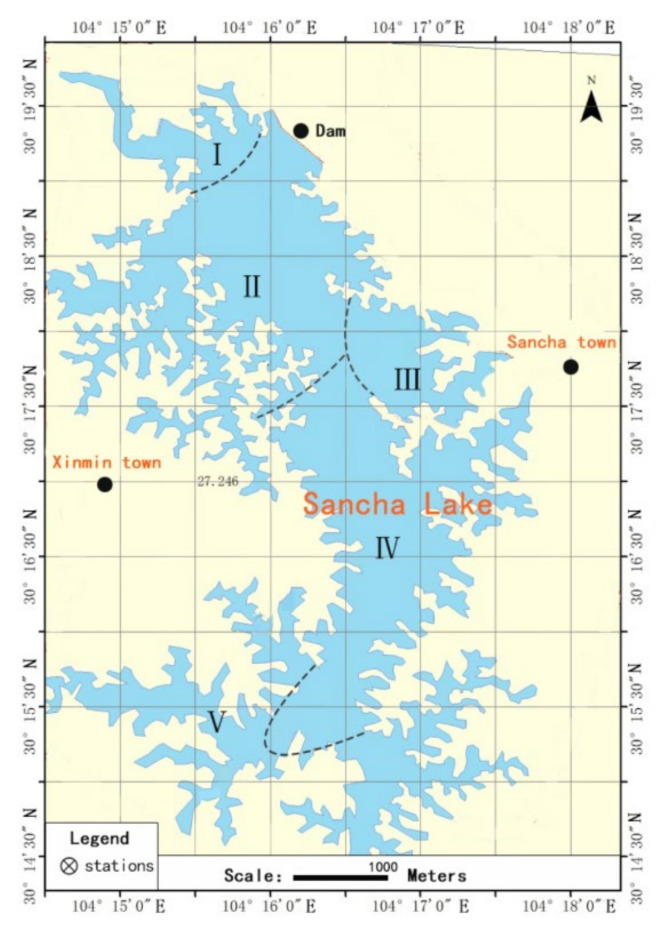
Sampling sites and function divisions of the Sancha Lake [15]. Note: (I) The main lake headwater area, (II) the highly concentrated area of the original cage culture, (III) the neighboring area with concentrated human activities, (IV) the relatively concentrated area of the pen culture, and (V) the reservoir tailwater area.

**Figure 2 ijerph-19-02320-f002:**
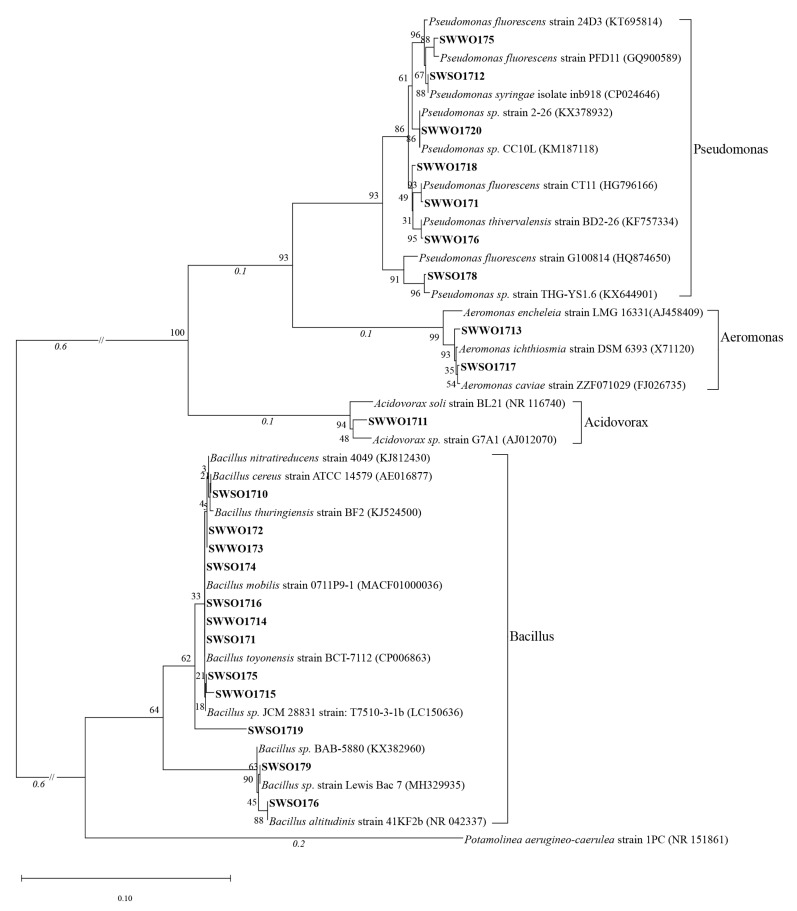
A neighbor-joining tree showing the phylogenetic relationships among 16S rDNA sequences of isolated OPB strains and their closely related sequences from EzBioCloud. The numbers at the nodes indicate the bootstrap values based on the neighbor-joining analyses of 1000 resampled datasets. The scale bar indicates evolutionary distance.

**Figure 3 ijerph-19-02320-f003:**
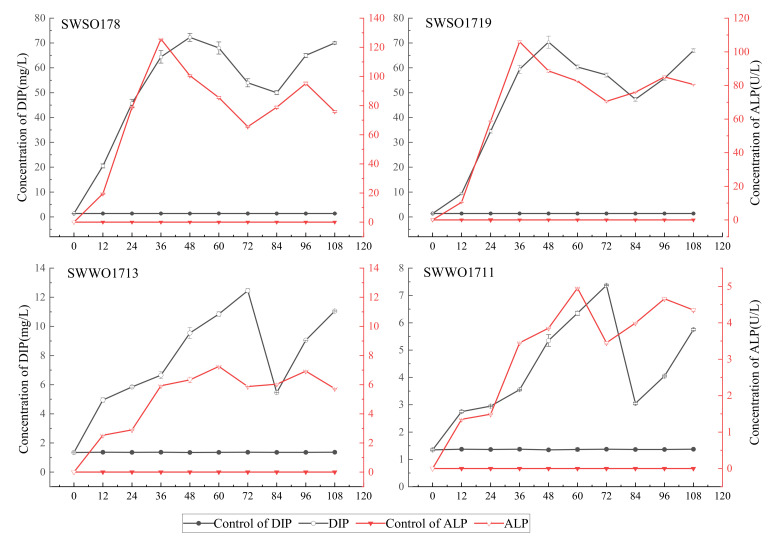
Changes in concentrations of ALP and DIP in SWSO178, SWSO1719, SWWO1713, and SWWO1711 liquid culture. Note: growing broth at 25 °C after 12, 24, 36, 48, 60, 72, 84, 96, and 108 h of growth. Each value is a mean of three independent replicates.

**Figure 4 ijerph-19-02320-f004:**
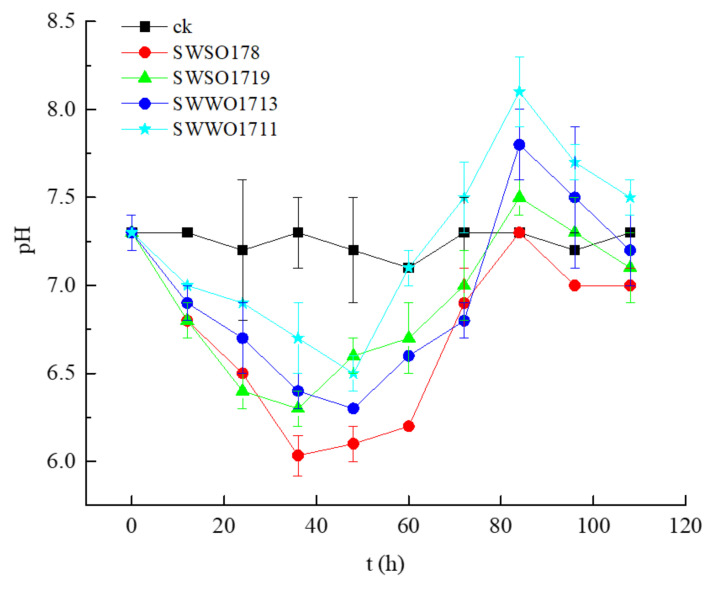
pH variation in the culture of four representative bacteria strains. Note: growing broth at 28 °C after 12, 24, 36, 48, 60, 72, 84, 96, and 108 h of growth. Each value is a mean of three independent replicates. Control: uninoculated control.

**Figure 5 ijerph-19-02320-f005:**
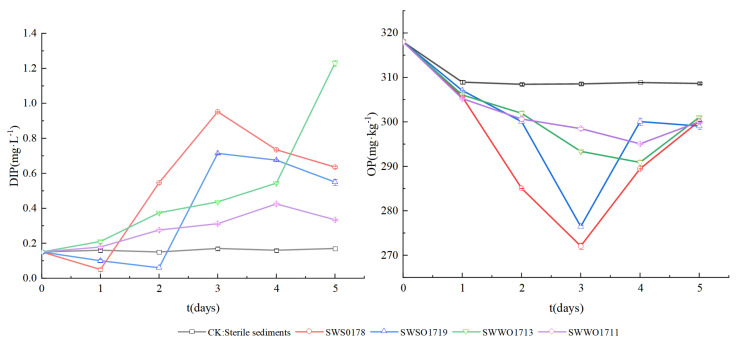
Solubilizing effect of four representative strains in sterilized sediments. Note: growing broth at 13 °C after 1, 2, 3, 4, and 5 days of growth. The DIP contents in the overlying water. The OP contents in the sediments. Each value is a mean of 3 independent replicates. CK: uninoculated control. The strains of SWSO178, SWSO1719, SWWO1713, and SWWO1711 indicate treatments inoculated, respectively.

**Table 1 ijerph-19-02320-t001:** Online BLAST results of 16S rDNA sequences for the 39 OPB strains in the sediments of Sancha Lake.

Bacteria Strain	Most Closely Related Strains (Accession Number) ^a^	Gene Identity (%) ^b^	Taxonomical Assignment	Query Length	Accession Number ^c^
SWSO171	*Bacillus thuringiensis* strain ATCC 10792(ACNF01000156)	100	*Bacillus thuringiensis*	1460	MK828267
SWSO172	*Bacillus altitudinis* strain 41KF2b (ASJC01000029)	99.93	*Bacillus altitudinis*	1456	MK828268
SWSO174	*Bacillus mobilis* strain 0711P9-1(MACF01000036)	100	*Bacillus mobilis*	1492	MK828270
SWSO175	*Bacillus proteolyticus* strain TD42(MACH01000033)	99.66	*Bacillus proteolyticus*	1475	MK828271
SWSO176	*Bacillus altitudinis* 41KF2b (ASJC01000029)	100	*Bacillus altitudinis*	1456	MK828272
SWSO177	*Bacillus stratosphericus* strain 41KF2a (ASJC01000029)	100	*Bacillus stratosphericus*	1472	MK828273
SWSO179	*Bacillus pumilus* strain ATCC 7061(ABRX01000007)	99.79	*Bacillus pumilus*	1457	MK828275
SWSO1710	*Bacillus pacificus* strain EB422 (KJ812450)	99.86	*Bacillus pacificus*	1473	MK828276
SWSO1711	*Bacillus altitudinis* 41KF2b (ASJC01000029)	100	*Bacillus altitudinis*	1469	MK828277
SWSO1713	*Bacillus thuringiensis* strain ATCC 10792(ACNF01000156)	100	*Bacillus thuringiensis*	1474	MK828278
SWSO1714	*Bacillus luti* strain TD41(MACI01000041)	99.80	*Bacillus luti*	1472	MK828279
SWSO1715	*Bacillus albus* strain N35-10-2(MAOE01000087)	99.86	*Bacillus albus*	1476	MK828280
SWSO1716	*Bacillus luti* strain TD41(MACI01000041)	100	*Bacillus luti*	1459	MK828281
SWSO1719	*Bacillus albus* strain N35-10-2(MAOE01000087)	96.52	*Bacillus jysancha* sp*.nov*	1464	MK863518
SWWO172	*Bacillus paramycoides* strain MCCC 1A04098(NR157734.1)	100	*Bacillus paramycoides*	1458	MK828283
SWWO173	*Bacillus nitratireducens* strain 4049(KJ812430)	99.93	*Bacillus nitratireducens*	1460	MK828284
SWWO178	*Bacillus altitudinis* 41KF2b(ASJC01000029)	100	*Bacillus altitudinis*	1469	MK828289
SWWO179	*Bacillus altitudinis* 41KF2b(ASJC01000029)	100	*Bacillus altitudinis*	1460	MK828290
SWWO1712	*Bacillus thuringiensis* strain ATCC 10792(ACNF01000156)	100	*Bacillus thuringiensis*	1469	MK828292
SWWO1714	*Bacillus toyonensis* strain BCT-7112 (CP006863)	100	*Bacillus toyonensis*	1460	MK828294
SWWO1715	*Bacillus* sp. strain AFS096926(NVLJ01000028)	99.58	*Bacillus* sp.	1418	MK828295
SWWO1716	*Bacillus mobilis* strain 0711P9-1(MACF01000036)	99.93	*Bacillus mobilis*	1472	MK828296
SWSO178	*Pseudomonas* sp. strain LY1(LSSW01000001)	97.93	*Pseudomonas* sp.	1413	MK828274
SWSO1712	*Pseudomonas fluorescens* strain PgKB31(MH553942.1)	99.51	*Pseudomonas fluorescens*	1443	MK834812
SWWO171	*Pseudomonas silesiensis* strain A3 (KX276592)	99.86	*Pseudomonas silesiensis*	1446	MK828282
SWWO174	*Pseudomonas* sp. strain R28 (CM002330)	100	*Pseudomonas* sp.	1443	MK828285
SWWO175	*Pseudomonas* sp. strain R28 (CM002330)	99.79	*Pseudomonas* sp.	1453	MK828286
SWWO176	*Pseudomonas kilonensis* strain ATCC 49054(EU391388)	99.38	*Pseudomonas kilonensis*	1450	MK828287
SWWO177	*Pseudomonas silesiensis* strain A3(KX276592)	100	*Pseudomonas silesiensis*	1459	MK828288
SWWO1710	*Pseudomonas baetica* strain a390(FM201274)	99.05	*Pseudomonas baetica*	1465	MK834811
SWWO1718	*Pseudomonas* sp. strain 11K1(CP035088)	100	*Pseudomonas* sp.	1469	MK863513
SWWO1720	*Pseudomonas yamanorum* strain 8H1(EU557337)	99.86	*Pseudomonas yamanorum*	1457	MK863515
SWWO1713	*Aeromonas veronii* strain CECT 4257(CDDK01000015)	99.44	*Aeromonas veronii*	1437	MK828293
SWSO1717	*Aeromonas encheleia* strain LMG 16331(AJ458409)	98.56	*Aeromonas encheleia*	1443	MK863516
SWSO1718	*Aeromonas encheleia* strain LMG 16331(AJ458409)	98.52	*Aeromonas encheleia*	1438	MK863517
SWWO1719	*Aeromonas veronii* strain CECT 4257(CDDK01000015)	99.83	*Aeromonas veronii*	1468	MK863514
SWWO1717	*Aeromonas veronii* strain CECT 4257(CDDK01000015)	98.63	*Aeromonas veronii*	1426	MK863512
SWWO1721	*Spirosoma* soli MIMBbqt12^T^(KT347096)	96.32	*Spirosoma lacussanchae* sp*.nov*	1459	KX580025
SWWO1711	*Acidovorax* sp. strain Root219(LMIJ01000041)	99.51	*Acidovorax* sp.	1434	MK828291

Note: a. Most closely related strains observed in EzBioCloud. b. Percentage of identity with EzBioCloud analysis. c. Query length: length of 16S rDNA sequenced.

**Table 2 ijerph-19-02320-t002:** Solubilizing characteristics of the 39 OPB strains in the sediments of the Sancha Lake. In the same column, data with different letters, such as a, b, and c, indicate significant differences, while data with the same letter indicated insignificant differences at 0.05 level. Data with letters such as ab were insignificantly different from both data with letter a and data with letter b.

Bacteria Strain	Phosphate Solubilizing Halo (HD/CD) ^ⅰ^	DIP (mg·L^−1^) ^ⅱ^	ALP (U·L^−1^) ^ⅲ^	Cell Density (log CFU mL^−1^) ^ⅳ^
Control	—	1.35 ± 0.16 k	0	0
SWSO1712	4.1 ± 0.3 ab	21.92 ± 0.35 f	8.8 ± 0.9 cd	5.10 f
SWSO1718	1.2 ± 0.1 h	2.26 ± 0.20 j	5.6 ± 0.4 fg	5.98 cd
SWSO1717	1.3 ± 0.1 g	10.36 ± 0.20 gh	12.3 ± 0.16 b	5.06 f
SWSO171	3.0 ± 0.3 d	51.67 ± 1.12 c	6.2 ± 0.5 f	6.75 b
SWSO172	4.3 ± 0.4 a	62.65 ± 1.20 b	5.0 ± 0.5 g	6.85 ab
SWSO173	4.1 ± 0.3 ab	51.53 ± 1.32 c	5.4 ± 0.4 g	6.19 c
SWSO174	4.2 ± 0.2 ab	62.12 ± 0.82 b	5.1 ± 0.3 g	7.39 a
SWSO175	4.1 ± 0.3 ab	32.02 ± 1.42 e	8.1 ± 0.7 d	6.27 c
SWSO176	4.0 ± 0.2 b	32.25 ± 1.20 e	8.2 ± 0.9 d	6.65 b
SWSO177	3.0 ± 0.2 d	31.36 ± 0.97 e	7.9 ± 1.0 d	6.93 b
SWSO178	2.3 ± 0.2 ef	72.26 ± 1.27 a	3.5 ± 0.2 h	7.14 a
SWSO179	4.2 ± 0.2 ab	37.23 ± 1.30 d	7.5 ± 0.6 cd	5.79 cd
SWSO1710	3.5 ± 0.2 c	31.75 ± 1.10 e	8.0 ± 0.6 d	6.71 b
SWSO1711	3.7 ± 0.3 bc	22.09 ± 0.35 f	8.9 ± 1.1 cd	6.16 c
SWSO1713	3.6 ± 0.2 c	21.82 ± 0.35 f	8.9 ± 0.6 cd	6.75 b
SWSO1714	1.4 ± 0.1 g	9.82 ± 0.35 gh	12.7 ± 1. 6 ab	5.99 cd
SWSO1715	1.8 ± 0.3 fg	3.78 ± 0.15 i	6.9 ± 0.6 e	6.09 cd
SWSO1716	2.8 ± 0.3 e	13.82 ± 0.15 g	10.2 ± 0.9 c	5.27 e
SWSO1719	4.5 ± 0.1a	70.36 ± 1.20 a	3.8 ± 0.2 h	7.16 a
SWWO171	3.2 ± 0.2 cd	21.78 ± 0.12 f	8.5 ± 0.6 cd	6.62 b
SWWO172	1.3 ± 0.1 g	2.74 ± 0.11 j	6.4 ± 0. 6 f	5.67 d
SWWO173	2.1 ± 0.1 f	3.67 ± 0.10 i	6.8 ± 0.9 e	6.02 cd
SWWO174	1.3 ± 0.3 g	12.27 ± 0.22 g	11.5 ± 1.6 bc	5.90 cd
SWWO175	ND	1.95 ± 0.09 jk	6.2 ± 0.2 f	5.06 f
SWWO176	3.1 ± 0.2 d	22.07 ± 0.38 f	8.5 ± 0.2 cd	6.16 c
SWWO177	2.1 ± 0.2 f	19.49 ± 0.05 f	9.3 ± 0.6 c	6.05 cd
SWWO178	3.1 ± 0.1d	30.5 ± 0.05 e	8.3 ± 0.7 d	5.67 d
SWWO1718	ND	50.56 ± 1.31 c	7.2 ± 0.6 e	6.99 b
SWWO1719	ND	2.41 ± 0.35 j	6.7 ± 0.7 e	5.82 cd
SWWO1720	ND	2.37 ± 0.35 j	6.2 ± 0.6 f	6.99 b
SWWO179	3.0 ± 0.2 d	27.52 ± 0.51 ef	8.1 ± 0.9 d	6.16 c
SWWO1710	1.3 ± 0.1 g	2.52 ± 0.08 j	6.0 ± 0.8 f	5.08 f
SWWO1711	1.1 ± 0.1 h	7.37 ± 0.2 h	13.7 ± 1.6 a	5.16 f
SWWO1712	2.0 ± 0.2 f	18.82 ± 0.18 fg	9.4 ± 1.1 c	5.29 d
SWWO1713	1.2 ± 0.1 h	12.45 ± 0.08 g	11.3 ± 1.6 bc	5.25 e
SWWO1714	1.2 ± 0.2 h	3.32 ± 0.35 i	6.5 ± 0.6 ef	4.81 g
SWWO1715	1.6 ± 0.4 fg	2.09 ± 0.35 j	5.3 ± 0.4 g	5.83 cd
SWWO1716	1.3 ± 0.2 g	1.92 ± 0.35 jk	4.9 ± 0.3 g	4.80 g
SWWO1717	1.5 ± 0.1 g	1.82 ± 0.35 jk	4.5 ± 0.5 g	4.87 fg
SWWO1721	1.2 ± 0.2 h	3.25 ± 0.12 i	10.4 ± 0.9 c	5.26 d

Note: Each value is represented by mean ± S.E; Control: no inoculation; ND: not detectable. i. HD/CD, the ratio of halo zone to colony diameter in the lecithin solid medium; ii. the maximum DIP value of sample after subtracting the CK, DIP value in the lecithin-contained liquid medium; iii. the enzyme activity of ALP (U) in which maximum levels of soluble P were released; iv. Colony-forming units at time of maximum levels of soluble P released by each bacterium.

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
