# Peer review of "Diversity and Phosphate Solubilizing Characteristics of Cultivable Organophosphorus-Mineralizing Bacteria in the Sediments of Sancha Lake"

_ijerph, 2022, doi:10.3390/ijerph19042320_

Round 1
Reviewer 1 Report
The paper has highly improved.
Three small comments.
Line 49 organo (not Organo)
Line 232 could (not Could)
After table 2 and before line 322 add space.
--
In real sediment, the anaerobic bacteria may have a higher role, but it should be studied in another research if you have a possibility for this work
Author Response
Comments and Suggestions for Author1
- Line 49 organo (not Organo)
Reply: Accepted and it was corrected.
- Line 232 could (not Could)
Reply: Accepted and it was corrected.
- After table 2 and before line 322 add space.
Reply: Accepted and it has been added.

Reviewer 2 Report
In this new submission, the authors have considered most of my previous comments, although there were not able to present results on the growth of the tested organophosphorus-mineralizing bacteria. Therefore, I suggest that discussion of stages of bacterial growth related to enzymatic activity and other parameters analyzed should be taken with caution (L318-L323) and make aware the readers of that.
In the caption to Fig. 3, ‘Ck’ is not necessary.
Author Response
Comments and Suggestions for Authors
2
- I suggest that discussion of stages of bacterial growth related to enzymatic activity and other parameters analyzed should be taken with caution (L318-L323) and make aware the readers of that.
Reply: Accepted and bacterial growth related to enzymatic activity and other parameters analyzed has been added n discussion of stages.
2) In the caption to Fig. 3, ‘Ck’ is not necessary.
Reply: Accepted and ‘Ck’ were deleted from the caption of Fig. 3.
